# Policy-Guided Causal State Representation for Offline Reinforcement Learning Recommendation

## Abstract

In offline reinforcement learning-based recommender systems (RLRS), learning effective state representations is crucial for capturing user preferences that directly impact long-term rewards. However, raw state representations often contain high-dimensional, noisy information and components that are not causally relevant to the reward. Additionally, missing transitions in offline data make it challenging to accurately identify features that are most relevant to user satisfaction. To address these challenges, we propose Policy-Guided Causal Representation (PGCR), a novel two-stage framework for causal feature selection and state representation learning in offline RLRS. In the first stage, we learn a causal feature selection policy that generates modified states by isolating and retaining only the causally relevant components (CRCs) while altering irrelevant components. This policy is guided by a reward function based on the Wasserstein distance, which measures the causal effect of state components on the reward and encourages the preservation of CRCs that directly influence user interests. In the second stage, we train an encoder to learn compact state representations by minimizing the mean squared error (MSE) loss between the latent representations of the original and modified states, ensuring that the representations focus on CRCs and filter out irrelevant variations. We provide a theoretical analysis proving the identifiability of causal effects from interventions, validating the ability of PGCR to isolate critical state components for decision-making. Extensive experiments demonstrate that PGCR significantly improves recommendation performance, confirming its effectiveness for offline RL-based recommender systems.

## Keywords

Offline Reinforcement Learning, Recommendation, Causal State Representation

## 1 Introduction

Reinforcement Learning (RL) has emerged as a powerful approach for developing recommender systems (RS), where the objective is to sequentially learn a policy that maximizes long-term rewards, typically measured by user satisfaction or engagement. Unlike traditional recommendation methods that primarily aim to optimize immediate rewards, RLRS focuses on learning a recommendation strategy that adapts to user preferences over time [5]. This allows RLRS to dynamically update recommendations based on user feedback, aiming to improve long-term outcomes and enhance user experiences.

However, deploying RL in recommender systems poses significant challenges. Traditional RLRS rely on continuous user interaction to learn and adapt their policies, which may be impractical in many real-world applications due to concerns such as exploration risks, privacy issues, and computational costs [2]. To address these challenges, offline RL-based recommender systems have been proposed, where the goal is to learn optimal recommendation policies from a fixed dataset of historical user interactions without further online data collection. This offline setting leverages existing data to refine and optimize recommendations, but it also introduces some challenges.

A critical aspect in offline RLRS is learning efficient state representations [1, 2]. In the offline setting, the agent must learn solely from historical data without additional interactions, making the challenges of high-dimensional and noisy state representations more pronounced. The state space, which includes information about user interactions, context, and preferences, is fundamental for deciding actions (i.e., recommendations). However, raw state representations are often complex and may contain components that are not causally relevant to the reward.

Recent advances in representation learning in RL have focused on extracting abstract features from high-dimensional data to enhance the efficiency and performance of RL algorithms [14, 15]. However, these challenges are compounded in the context of offline RLRS due to the static nature of the data and the inability to interact with the environment. Techniques such as those developed by Zhang et al. [31], which use the bisimulation metric to learn representations that ignore task-irrelevant information, may encounter challenges when applied directly to offline settings. In particular, missing transitions in the offline dataset can particularly impair the effectiveness of the bisimulation principle, resulting in inaccurate state representation and poor estimation [30]. Moreover, the complexity and high-dimensional nature of user data in offline RLRS require isolating the components that are causally relevant to the reward, rather than merely compressing the state space. Thus, there is a need for targeted techniques that emphasize causally critical state components within the constraints of offline learning.

To address these challenges, we propose a policy-guided approach for causal feature selection and state representation learning. Our approach is designed to use a policy to generate intervened states that isolate and retain only the causally relevant components (CRCs). By focusing on the features that directly impact user satisfaction, this method enables the state representation to concentrate on the most informative components, reducing noise and irrelevant variations. Additionally, by creating targeted interventions, this approach augments offline datasets, enhancing the learning of state representations even with finite datasets.

We introduce a method called PGCR (Policy-Guided Causal Representation), which operates in two stages. In the first stage, we learn a causal feature selection policy that generates modified states, retaining the CRCs and modifying the causally irrelevant components (CIRCs). We quantify the causal effect of the state components on the reward, which reflects user feedback, by using the Wasserstein distance between the original and modified reward distributions. This metric effectively measures the distributional

change caused by the interventions, and we use it to design a reward function that encourages the retention of CRCs while altering CIRCs. Furthermore, we provide theoretical analysis on the identifiability of the causal effects resulting from these interventions. In the second stage, we leverage the learned causal feature selection policy to guide the training of a state representation encoder. Given a pair consisting of an original state and its modified counterpart generated by the causal feature selection policy, the encoder is trained to produce latent representations that preserve only the CRCs. Specifically, we minimize the mean squared error (MSE) loss between the latent representations of the original and modified states, encouraging the encoder to ignore irrelevant variations and focus on causally meaningful features. This process allows the encoder to map states into a latent space where only the information necessary for optimal decision-making is preserved.

Our contributions are as follows:

- We propose PGCR, a novel two-stage framework for offline RL-based recommender systems. In the first stage, we learn a causal feature selection policy to generate modified states that retain causally relevant components. In the second stage, we train an encoder to learn state representations concentrating on these components.
- We design a reward function based on the Wasserstein distance to guide the causal feature selection policy in identifying and retaining the state components that directly influence user interests.
- We provide a theoretical analysis proving the identifiability of causal effects from interventions, ensuring that our method isolates the components of the state critical for decision-making.
- Extensive experiments demonstrate the effectiveness of PGCR in improving recommendation performance in offline RL-based recommender systems.

## 2 Preliminaries

### 2.1 Offline RL-Based Recommender Systems

Offline Reinforcement Learning (RL) in Recommender Systems (RS) aims to optimize decision-making by learning solely from historical user interaction data within the framework of a Markov Decision Process (MDP). The MDP is represented by the tuple $\langle \mathcal{S}, \mathcal{A}, \mathcal{R}, \mathcal{P}, \gamma \rangle$, where:

- $\mathcal{S}$ represents the state space, encompassing user data, historical interactions, item characteristics, and contextual factors.
- $\mathcal{A}$ denotes the action space, which includes all candidate items available for recommendation.
- $\mathcal{R} : \mathcal{S} \times \mathcal{A} \to \mathbb{R}$ defines the reward function, based on user feedback such as clicks, ratings, or engagement metrics.
- $\mathcal{P}$ describes the transition probabilities, governing the dynamics of state transitions.
- $\gamma$ is the discount factor, used to balance immediate and future rewards.

Unlike online RL, the agent does not interact with the environment in real-time but must infer the optimal policy solely from historical data. In this MDP setup, the agent (RS) learns from a fixed dataset $\mathcal{D}$ of interactions collected by a behavior policy. Each entry in this dataset consists of a state $s_t$, an action $a_t$ taken by the behavior policy, the resulting reward $r_t$, and the next state $s_{t+1}$. The primary goal of the RS agent is to learn a policy $\pi : \mathcal{S} \to \mathcal{A}$ that maximizes the cumulative discounted return, thereby ensuring the long-term effectiveness of the recommendations provided to the user.

### 2.2 Causal Models

Causal models provide a structured way to represent and analyze the causal relationships among a set of variables. Consider a finite set of random variables denoted by $\mathbf{X} = \{X_1, X_2, \ldots, X_n\}$, each associated with an index in $\mathbf{V} = \{1, 2, \ldots, n\}$. These variables have a joint distribution $P_{\mathbf{X}}$ and a joint density function $p(\mathbf{x})$. A causal graphical model is represented by a Directed Acyclic Graph (DAG) $\mathcal{G} = (\mathbf{V}, \mathcal{E})$, where $\mathbf{V}$ is the set of nodes, each corresponding to one of the variables in $\mathbf{X}$ and $\mathcal{E}$ is the set of directed edges between the nodes, indicating direct causal influences.

**Definition 2.1** (Structural Causal Model). A Structural Causal Model (SCM) $\mathcal{M} = (\mathbf{S}, P_{\mathbf{U}})$ associated with a DAG $\mathcal{G}$ consists of a set $\mathbf{S}$ of structural equations:

$$X_i = f_i(\text{PA}_i, U_i), \quad i = 1, 2, \ldots, n,$$

where $\text{PA}_i \subseteq \mathbf{X} \setminus \{X_i\}$ denotes the set of parent variables (direct causes) of $X_i$ in the graph $\mathcal{G}$. $U_i$ represents the exogenous (noise) variables, accounting for unobserved factors, and $\mathbf{U} = \{U_1, \ldots, U_n\}$ is the set of all such variables. A joint distribution $P_{\mathbf{U}}$ over the noise variables $\mathbf{U}$, assumed to be jointly independent.

Each structural function $f_i$ specifies how $X_i$ is generated from its parents $\text{PA}_i$ and the noise term $U_i$. The combination of the structural equations $\mathbf{S}$ and the distribution $P_{\mathbf{U}}$ induces a joint distribution $P_{\mathbf{X}}$ over the endogenous variables $\mathbf{X}$.

**Definition 2.2** (Intervention). An intervention in an SCM $\mathcal{M}$ is an operation that modifies one or more of the structural equations in $\mathbf{S}$. Specifically, suppose we replace the structural equation for variable $X_j$ with a new equation:

$$X_j = \hat{f}_j(\widehat{\text{PA}}_j, \hat{U}_j).$$

This results in a new SCM $\hat{\mathcal{M}}$, reflecting the intervention on $X_j$. The corresponding distribution changes from the observational distribution $P_{\mathbf{X}}^{\mathcal{M}}$ to the interventional distribution $P_{\mathbf{X}}^{\hat{\mathcal{M}}}$, expressed as:

$$P_{\mathbf{X}}^{\hat{\mathcal{M}}} = P_{\mathbf{X}}^{\mathcal{M}; \, do(X_j = \hat{f}_j(\widehat{\text{PA}}_j, \hat{U}_j))},$$

where the *do*-operator $do(X_j = \hat{f}_j(\widehat{\text{PA}}_j, \hat{U}_j))$ denotes the intervention that replaces the structural equation for $X_j$.

## 3 Methodology

### 3.1 Problem Formulation

To learn a policy that identifies the causally relevant components in the state, we first represent the MDP from a causal modeling perspective. Assuming there are no unobserved confounders, the SCMs for the MDP can be formulated using deterministic equations augmented with exogenous noise variables to capture stochasticity, as shown in Figure 1 (a):

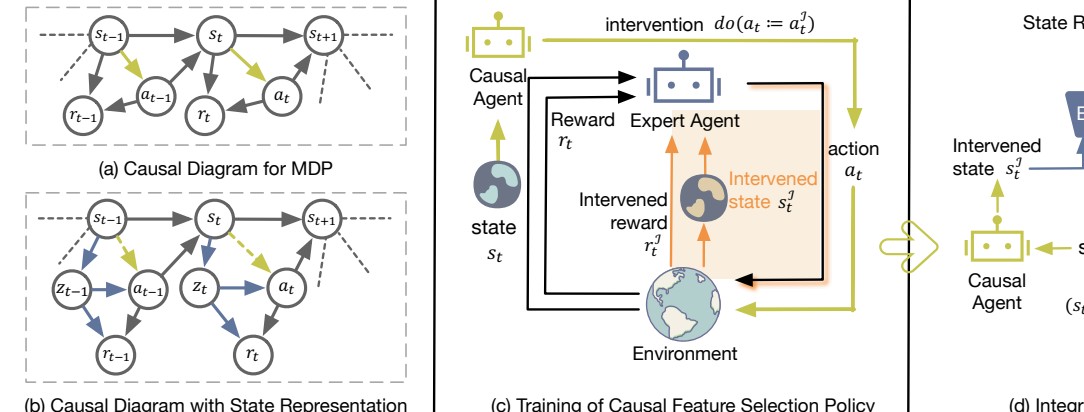

(a) Causal Diagram for MDP

(b) Causal Diagram with State Representation

(c) Training of Causal Feature Selection Policy

(d) Integrated Learning Process

Figure 1: (a) A graphical representation showing the causal relationships between state $s_t$, action $a_t$, and reward $r_t$ over time, with green lines indicating where the causal agent makes interventions. (b) An extended diagram incorporating the latent state $z_t$ (blue lines), illustrating that actions $a_t$ depend on $z_{t-1}$ rather than $s_{t-1}$ (green, dashed lines) as described in Proposition 2. (c) The causal agent intervenes on actions to generate modified states $s_t^{\mathcal{I}}$, while the expert agent collects rewards from both original and modified states to train the causal policy. (d) The causal agent uses the offline dataset to generate modified states, which the encoder processes to learn latent representations for training the recommendation agent.

$$s_{t+1} = f_P(s_t, a_t, \epsilon_{t+1}), \quad a_t = \pi_t(s_t, \eta_t), \quad r_t = f_R(s_t, a_t). \quad (1)$$

In this formulation, the state transition function $f_P$ determines the next state $s_{t+1}$ based on the current state $s_t$, action $a_t$, and exogenous noise $\epsilon_{t+1}$. The policy function $\pi_t$ selects the action $a_t$ given the current state $s_t$ and exogenous noise $\eta_t$. The reward function $f_R$ assigns a reward $r_t$ based on the current state $s_t$ and action $a_t$.

By modeling the MDP in this way, we can explicitly analyze how different components of the state causally affect rewards, allowing us to focus on the elements of $s_t$ that have a direct causal impact on $r_t$. To differentiate their levels of influence on learning user interest representations, we decompose the state $s_t \in \mathcal{S}$ into two disjoint components: Causally Relevant Components (CRCs) and Causally Irrelevant Components (CIRCs).

Since the rewards in a recommender system reflect users' interests, we measure the causal effect on user preferences through the rewards. Formally, the CRCs are identified as parts of the state that contain critical information about the user's interest. Modifications to the CRCs lead to significant changes in rewards and the items recommended. In contrast, CIRCs are state components that have minimal influence on representing user interests, so altering them has a weak causal effect on rewards in the SCMs. Given the distinction between these components, the core of our approach is to learn a policy that can accurately identify and retain the causally relevant components of a state.

## 3.2 Causal Feature Selection Policy

Given a tuple $\{s_t, a_t, s_{t+1}, r_t\}$, the causal feature selection policy identifies the CRCs by making the atomic intervention on the action $a_t$, which is forced to take on some value $a_t^{\mathcal{I}}$. Formally, this atomic

intervention, which we denote by $do(a_t := a_t^{\mathcal{I}})$, or $do(a_t)$ for short, amounts to removing the equation $a_t = \pi_t(s_t, \eta_t)$ from the model and substituting $a_t := a_t^{\mathcal{I}}$ in the remaining equations. The new model thus created represents the system's behavior under the intervention $do(a_t := a_t^{\mathcal{I}})$ and, when solved for the distribution of $s_{t+1}$, yields the causal effect of $a_t$ on $s_{t+1}$, which is denoted $P^{do(a_t := a_t^{\mathcal{I}})}(s_{t+1})$.

Your proposition looks logically sound, and the structure is clear. To improve consistency, here's a refined version of the proposition with a small clarification at the end:

**Proposition 1** (Identifiability). *Suppose the state $s_t$ and action $a_t$ are observable and form an MDP, as described in Equation (7). The variable $s_t$ satisfies the back-door criterion (see Appendix A) relative to the pair of variables $(a_t, s_{t+1})$ because it meets the following criteria: There is no descendant of $a_t$ in $s_t$, and all paths between $a_t$ and $s_{t+1}$ that contain an arrow into $a_t$ are blocked by $s_t$. Therefore, the causal effect of $a_t$ on $s_{t+1}$ is identifiable.*

The proof of Proposition 1 is given in Appendix B. Consequently, the probability distribution for the state $s_{t+1}$ induced after intervention is given by the formula:

$$
\begin{aligned}
&P^{\mathcal{M}; do(a_t := a_t^{\mathcal{I}})}(s_{t+1}) \\
&= \sum_{s_t} \int_{\epsilon_{t+1}} P(s_{t+1} \mid do(a_t), s_t, \epsilon_{t+1}) \, P(\epsilon_{t+1}) \, P(s_t \mid do(a_t)) \, d\epsilon_{t+1} \\
&= \sum_{s_t} \int_{\epsilon_{t+1}} P(s_{t+1} \mid s_t, a_t^{\mathcal{I}}, \epsilon_{t+1}) \, P(\epsilon_{t+1}) \, P(s_t) \, d\epsilon_{t+1} \\
&= \mathbb{E}_{s_t, \epsilon_{t+1}} \left[ P(s_{t+1} \mid s_t, a_t^{\mathcal{I}}, \epsilon_{t+1}) \right].
\end{aligned}
\tag{2}
$$

After the causal feature selection policy intervenes on the action $a_t$, setting it to a specific value $a_t^{\mathcal{I}}$, the environment transitions to a new state $s^{\mathcal{I}}$. This intervened state $s^{\mathcal{I}}$ is expected to preserve only the causally relevant components of the original state $s_t$, while any causally irrelevant CIRCs are modified or filtered out.

Since the CRCs are the parts of $s_t$ that have a significant causal impact on rewards, we regard the new state $s^{\mathcal{I}}$, induced by the intervention on $a_t$, as an effective intervention on $s_t$ in the original tuple $\{s_t, a_t, s_{t+1}, r_t\}$. By comparing the rewards obtained before and after the intervention, we can evaluate the causal effect of the original state $s_t$ on the reward $r_t$, isolating the impact of the causally relevant components.

Formally, following Pearl's rules of *do*-calculus [21], as outlined in Appendix A, the causal effect of $s_t$ on $r_t$ is given by the formula:

$$
\begin{aligned}
& P^{\mathcal{M}; do(s_t := s^{\mathcal{I}})}(r_t) \\
&= \sum_{a_t} \int_{\eta_t} P(r_t \mid do(s_t := s^{\mathcal{I}}), a_t, \eta_t)\, P(\eta_t)\, P(a_t \mid do(s_t := s^{\mathcal{I}}))\, d\eta_t \\
&= \sum_{a_t} \int_{\eta_t} P(r_t \mid s_t := s^{\mathcal{I}}, a_t, \eta_t)\, P(\eta_t)\, P(a_t \mid s_t := s^{\mathcal{I}})\, d\eta_t \\
&= \mathbb{E}_{a_t, \eta_t} \left[ P(r_t \mid s_t := s^{\mathcal{I}}, a_t, \eta_t) \right].
\end{aligned}
\tag{3}
$$

If the intervened probability distribution of the reward is similar to the original distribution, substituting $s_t$ with $s^{\mathcal{I}}$ has a minor causal effect on the reward. This indicates that the causally CRCs of $s_t$ that significantly influence learning the user's interest have been retained. To quantify this effect, we measure the distance between the two probability distributions of the reward before and after the intervention. Inspired by bisimulation for state abstraction [7], we adopt the first-order Wasserstein distance to measure how the intervened reward probability distribution $P^{\mathcal{M}; do(s_t := s^{\mathcal{I}})}(r_t)$ differs from the original distribution $P^{\mathcal{M}}(r_t)$:

$$
W_1 \left( P^{\mathcal{M}; do(s_t := s^{\mathcal{I}})}(r_t),\, P^{\mathcal{M}}(r_t) \right) = \inf_{\gamma \in \Gamma(P^{\mathcal{I}}, P^{\mathcal{M}})} \int_{\mathcal{R} \times \mathcal{R}} |r - r'|\, d\gamma(r, r'),
\tag{4}
$$

where $\Gamma\left(P^{\mathcal{I}}, P^{\mathcal{M}}\right)$ is the set of all joint distributions $\gamma(r, r')$ with marginals $P^{\mathcal{M}; do(s_t := s^{\mathcal{I}})}(r_t)$ and $P^{\mathcal{M}}(r_t)$. A small Wasserstein distance indicates that the intervention on the state $s_t$ has a negligible effect on the reward distribution, suggesting that the components altered by the intervention are causally irrelevant to the reward. Conversely, a large Wasserstein distance implies that the intervention significantly changes the reward distribution, highlighting the causal relevance of the components modified in the state.

By evaluating the Wasserstein distance between the original and intervened reward distributions, we can quantify the causal effect of the state components on the reward. This measurement not only guides the causal feature selection policy in identifying and retaining the causally relevant components in the state but also serves as a crucial guide for the agent's learning process. To operationalize this measurement within the agent's learning, we introduce an effective reward function defined as:

$$
r_t = \exp \left( -\lambda\, W_1 \left( P^{\mathcal{M}; do(s_t := s^{\mathcal{I}})}(r_t),\, P^{\mathcal{M}}(r_t) \right) \right),
\tag{5}
$$

where $\lambda \in (0, 1]$ is a scaling parameter that controls the sensitivity of the reward to changes in the Wasserstein distance.

By maximizing this reward, the agent is incentivized to select actions that minimize the Wasserstein distance between the intervened and original reward distributions. This encourages the agent to choose actions that retain the causally relevant components of the state, effectively filtering out causally irrelevant features. Consequently, the agent adjusts its policy to focus on the essential causal elements.

### 3.3 Policy-Guided State Representation

Having identified the CRCs of the state through our causal feature selection policy, we proceed to learn a state representation that effectively captures these essential components. The objective is to encode the current state $s_t$ and its intervened counterpart $s_t^{\mathcal{I}}$ into a latent space where only the CRCs are preserved, and the CIRCs are minimized or disregarded. To achieve this, we employ an encoder trained using mean squared error (MSE) loss, which focuses on aligning the representations of $s_t$ and $s_t^{\mathcal{I}}$ by minimizing the differences in their latent representations.

By using the causal feature selection policy to generate modified states $s_t^{\mathcal{I}}$, which share the same CRCs but differ in CIRCs compared to the original state $s_t$, we provide the encoder with pairs of states that should be mapped to similar latent representations. The MSE loss between the latent representations of $s_t$ and $s_t^{\mathcal{I}}$ encourages the encoder to focus on the CRCs and ignore the CIRCs. Moreover, generating modified states $s_t^{\mathcal{I}}$ through interventions allows us to augment the dataset, addressing the issue of missing transitions commonly encountered in offline recommender systems.

Practically, we design an encoder network $\phi$ that processes the input states and outputs their latent representations:

$$
z_t = \phi(s_t), \quad z_t^{\mathcal{I}} = \phi(s_t^{\mathcal{I}}).
$$

We train the encoder by minimizing the mean squared error (MSE) loss between the latent representations of $s_t$ and $s_t^{\mathcal{I}}$:

$$
J = \| \phi(s_t) - \phi(s_t^{\mathcal{I}}) \|_2^2.
\tag{6}
$$

This loss function encourages the encoder to focus on the CRCs by reducing the differences in the latent representations of $s_t$ and $s_t^{\mathcal{I}}$, which differ only in their CIRCs.

**Proposition 2** (Optimal Policy Based on Latent State Representation). Let $s_t \in \mathcal{S}$ be the full state at time $t$, and let $G = \sum_{k=0}^{\infty} \gamma^k r_{t+k}$ be the expected discounted return. Let $\phi : \mathcal{S} \to \mathcal{Z}$ be an encoder that maps $s_t$ to a latent state representation $z_t = \phi(s_t) \in \mathcal{Z}$, capturing the causally relevant components. Suppose for $z_t$, we have:

- $r_t \perp\!\!\!\perp s_t \mid z_t, a_t$.
- For all $s_{t-1}, s_{t-1}^{\circ} \in \mathcal{S}$ with $\phi(s_{t-1}) = \phi(s_{t-1}^{\circ})$, $p(\phi(s_t) \mid s_{t-1}) = p(\phi(s_t) \mid s_{t-1}^{\circ})$.

Then the optimal policy $\pi_{\text{opt}}$ depends only on the latent state representation $z_t$, and not on the full state $s_t$. That is, there exists

$$
\pi_{\text{opt}} \in \arg \max_{\pi} \mathbb{E}[G],
$$

such that

$$\pi_{\text{opt}}(a_t \mid s_{t-1}) = \pi_{\text{opt}}(a_t \mid s_{t-1}^{\circ}) \quad \forall s_{t-1}, s_{t-1}^{\circ} : \phi(s_{t-1}) = \phi(s_{t-1}^{\circ}).$$

The proof of Proposition 2 is given in Appendix C. This proposition shows that using the encoder $\phi(s_t)$ as a means of simplifying the state is theoretically justified. The encoder learns to isolate the CRCs from the full state, ensuring that the resulting latent representation $z_t$ contains all information needed for decision-making. This supports the approach of training an encoder to map states into a latent space that focuses on the essential causal features.

---

**Algorithm 1:** Training Procedure for Causal Feature Selection Policy

---

**Input:** Initial parameters $\theta_{\mu^c}$, $\theta_{\phi^c}$; replay buffer $D_c$; reward buffers $R, \hat{R}$

**for** $episode = 1$ to $E$ **do**

    **for** $t = 1$ to $T$ **do**

        Expert observes state $s_t$, executes action $a_t$, and stores reward $r_t$ in $R$;

        Causal agent intervenes with action $a_t^{\mathcal{I}}$ and obtains modified state $s_t^{\mathcal{I}}$;

        Expert observes $s_t^{\mathcal{I}}$, executes action $a_t$, and stores reward $\hat{r}_t$ in $\hat{R}$;

        Calculate reward $r$ based on the reward function ;

        `// See Eq. (5)`

        Store transition $(s_t, a_t^{\mathcal{I}}, s_t^{\mathcal{I}}, r)$ in replay buffer $D_c$;

        Sample minibatch from $D_c$ and update parameters $\theta_{\mu^c}$, $\theta_{\phi^c}$;

---

*3.3.1 Learning of Causal Feature Selection Policy.* The causal feature selection policy is trained by leveraging the reward function in Equation (5). The objective is to design interventions that retain the CRCs while minimizing changes to the reward distribution, thereby preserving the essential components influencing user satisfaction. The algorithm for learning the causal feature selection policy is provided in Algorithm 1.

A one-step illustration of the training process is depicted in Figure 1 (c). The causal feature selection policy is trained with the assistance of a pre-trained expert policy, which uses external knowledge to obtain both the observational and intervened reward distributions. The expert policy can be learned using any RL-based algorithm, and the causal feature selection policy can follow a similar approach.

During training, the expert policy interacts with the environment to collect tuples of the form $(s_t, a_t, r_t, s_{t+1})$, where $r_t$ contributes to the observational reward distribution. Simultaneously, the causal feature selection policy observes the state $s_t$ and intervenes on the action to generate a modified state $s_t^{\mathcal{I}}$. This modified state $s_t^{\mathcal{I}}$ is treated as an intervention on the original tuple's state. The expert policy then observes $s_t^{\mathcal{I}}$ and executes the original action $a_t$, thereby obtaining an intervened reward, which is used to construct the intervened reward distribution.

By maximizing the reward in Equation (5), the causal feature selection policy is incentivized to produce modified states $s_t^{\mathcal{I}}$ that yield reward distributions similar to the original. This similarity indicates that the CRCs are effectively retained while the CIRCs are altered, ensuring that the modified states preserve the key causal components.

*3.3.2 Integrated Learning Process.* In the offline RL setting, we integrate the causal feature selection policy with the training of both the state representation encoder and the recommendation policy, as depicted in Figure 1 (d). Given a current state $s_t$ from the offline dataset, the causal feature selection policy generates a modified state $s_t^{\mathcal{I}}$ that retains only the CRCs. The state pair $(s_t, s_t^{\mathcal{I}})$ is then used to train the encoder network $\phi$, which processes the input states and outputs their latent representations.

The encoder is trained by minimizing the loss defined in Equation (6), which encourages it to focus on the CRCs by reducing the differences in the latent representations of the state pairs, which differ only in their CIRCs. Consequently, the encoder learns to map states into a latent space where only the causally relevant features are preserved, effectively filtering out irrelevant variations.

The recommendation policy $\pi_{\text{Re}}$ is subsequently trained using the latent representations $z_t$ as inputs. Because the encoder prioritizes the CRCs, the recommendation policy is equipped to make decisions based on the most pertinent information influencing user satisfaction. The full algorithm for the integrated learning process is presented in Algorithm 2.

---

**Algorithm 2:** Integrated Learning Process

---

**Input:** Offline dataset $\mathcal{D}$; causal policy $\pi_C$; encoder $\phi$ with parameters initial $\theta$; recommendation policy $\pi_{\text{Re}}$ with initial parameters $\phi$; learning rate $\alpha$

**foreach** *training epoch* **do**

    **foreach** *batch* $\mathcal{B}$ *from* $\mathcal{D}$ **do**

        `// Generate Modified State Using Causal Feature Selection`

        **foreach** $(s_t, a_t, r_t, s_{t+1}) \in \mathcal{B}$ **do**

            Generate modified state $s_t^{\mathcal{I}} = \pi_C(s_t)$;

        `// Train Encoder Using MSE Loss`

        Encode states: $z_t = \phi(s_t)$, $z_t^{\mathcal{I}} = \phi(s_t^{\mathcal{I}})$;

        Compute MSE loss: $\mathcal{L}_{\text{encoder}} = \|z_t - z_t^{\mathcal{I}}\|_2^2$;

        Update encoder parameters: $\theta = \theta - \alpha \nabla_\theta \mathcal{L}_{\text{encoder}}$;

        `// Train Recommendation Policy Using Latent Representations`

        Update policy parameters $\phi$ with offline RL algorithm;

---

## 4 Experiments

In this section, we begin by performing experiments on an online simulator and recommendation datasets to highlight the remarkable performance of our methods. We then conduct an ablation study to demonstrate the effectiveness of the causal-indispensable state representation.

## 4.1 Experimental Setup

We introduce the experimental settings with regard to environments and state-of-the-art RL methods.

*4.1.1 Recommendation Environments.* For offline evaluation, we use the following benchmark datasets:

- **MovieLens-1M**[1]: These datasets, derived from the Movie-Lens website, feature user ratings of movies. The ratings are on a 5-star scale, with each user providing at least 20 ratings. Movies and users are characterized by 23 and 5 features, respectively.
- **Coat** [23]: is a widely used dataset that is proposed for product recommendation.
- **YahooR3**: a music recommendation dataset that proposed by [20].
- **KuaiRec**: a video recommendation dataset that proposed by [10] which is fully-observable.
- **KuaiRand**: a video recommendation dataset similar to KuaiRec but with a randomly exposed mechanism [11].

When converting them into the RL environments, we use the GRU as the state encoder for those offline datasets. In addition to those offline datasets, we also conduct experiments on an online simulation platform - VirtualTB [24],

*4.1.2 Baseline.* Since limited work focuses on causal state representation learning for offline RLRS, we selected the traditional RL algorithm as the baseline. In concurrent work, CIDS [27] proposes using conditional mutual information to isolate crucial state variables. The key difference between our work and CIDS is that CIDS is tailored for online RLRS, focusing primarily on the causal relationship between action and state. In contrast, our work addresses offline RLRS, incorporating the reward into the framework to train a policy that guides the learning of state representations. In our experiments, we employ the following algorithms as the baseline:

- **Deep Deterministic Policy Gradient (DDPG) [18]**: An off-policy method suitable for environments with continuous action spaces, employing a target policy network for action computation.
- **Soft Actor-Critic (SAC) [12]** : An off-policy maximum entropy Deep RL approach, optimizing a stochastic policy with clipped double-Q method and entropy regularization.
- **Twin Delayed DDPG (TD3) [8]**: An enhancement over DDPG, incorporating dual Q-functions, less frequent policy updates, and noise addition to target actions.

To evaluate the performance of the proposed PGCR, we have plugged the PGCR into those mentioned baselines to evaluate the performance.

*4.1.3 Evaluation Measures.* Following the previous work [29], we will use the cumulative reward, average reward and interaction length as the main evaluation metric for those mentioned offline datasets. For VritualTB, we use the embedded CTR as the main evaluation metric.

[1]https://grouplens.org/datasets/movielens/

## 4.2 Implementation Details

In our experiments, we first need to train the causal agent to conduct the intervention and thus generate the intervened state. The offline demonstration is required to train the causal agent. We use a DDPG algorithm to conduct the process to obtain the offline demonstrations for various datasets. The algorithm is trained for 100,000 timesteps, and we save the policy with the best performance during the evaluation stage. The saved policy will be used to generate the offline demonstrations. For the training of our proposed method, we set the learning rate to $10^{-4}$ for the actor-network and $10^{-3}$ for the critic network. The discount factor $\gamma$ is set to 0.95, and we use a soft target update rate $\tau$ of 0.001. The hidden size of the network is set to 128, and the replay buffer size is set to $10^6$.

For those baselines, we are using the standard hyper-parameters settings from the Tianshou[2].

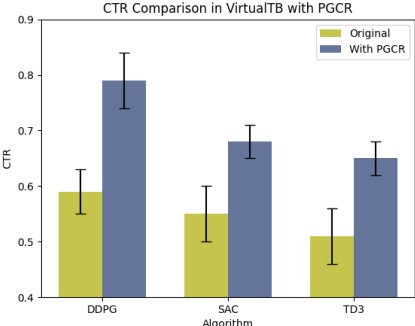

**Figure 2: The 1-step CTR performance in the VirtualTaobao simulation is presented as the mean with error bars.**

## 4.3 Overall Results

The results in Table 1 show that PGCR, a causal state representation learning method, significantly enhances state representation in reinforcement learning algorithms. Across different datasets, the PGCR-enhanced versions of standard algorithms (DDPG, SAC, TD3) demonstrate consistent improvements in cumulative and average rewards. This suggests that PGCR effectively strengthens the algorithms' ability to learn better state representations, leading to more informed decision-making and improved policy performance.

Moreover, the enhanced state representation provided by PGCR does not adversely affect the interaction length, which remains stable or slightly increases, indicating efficient learning processes. Additionally, the relatively low variance in the results for PGCR-enhanced methods further emphasizes their stability and reliability across different environments. These findings highlight the effectiveness of PGCR in boosting the overall learning and performance of reinforcement learning models by focusing on improved causal state representations.

## 4.4 Ablation Study

In this section, we aim to investigate the impact of the proposed causal agent on the final performance. To do this, we replaced the

[2]https://github.com/thu-ml/tianshou

**Table 1: Performance comparisons of our method with baselines on the MovieLens, Coat, KuaiRec and KuaiRand. The best results are highlighted in bold. The variance is also reported.**

| | MovieLens-1M | | | Coat | | |
|---|---|---|---|---|---|---|
| | Cumulative Reward | Average Reward | Interaction Length | Cumulative Reward | Average Reward | Interaction Length |
| DDPG | $9.3706 \pm 4.49$ | $3.0329 \pm 1.44$ | $3.11 \pm 0.02$ | $16.3348 \pm 7.23$ | $2.3277 \pm 1.03$ | $7.02 \pm 0.03$ |
| PGCR-DDPG | $\mathbf{13.0722 \pm 3.55}$ | $\mathbf{4.0587 \pm 1.10}$ | $\mathbf{3.22 \pm 0.03}$ | $\mathbf{19.4281 \pm 4.01}$ | $\mathbf{2.7675 \pm 0.57}$ | $7.02 \pm 0.05$ |
| SAC | $10.2424 \pm 3.66$ | $2.8852 \pm 1.03$ | $3.55 \pm 0.03$ | $17.5432 \pm 7.22$ | $2.4231 \pm 1.00$ | $7.24 \pm 0.02$ |
| PGCR-SAC | $\mathbf{13.4522 \pm 3.77}$ | $\mathbf{4.4544 \pm 1.25}$ | $\mathbf{3.02 \pm 0.05}$ | $\mathbf{20.4272 \pm 4.70}$ | $\mathbf{2.7164 \pm 0.63}$ | $\mathbf{7.52 \pm 0.10}$ |
| TD3 | $10.1620 \pm 4.90$ | $2.9410 \pm 1.42$ | $3.45 \pm 0.02$ | $16.3232 \pm 7.02$ | $2.3542 \pm 1.01$ | $6.93 \pm 0.03$ |
| PGCR-TD3 | $\mathbf{14.1281 \pm 5.21}$ | $\mathbf{3.4375 \pm 1.27}$ | $\mathbf{4.11 \pm 0.02}$ | $\mathbf{19.1192 \pm 3.81}$ | $\mathbf{2.5323 \pm 0.50}$ | $\mathbf{7.55 \pm 0.11}$ |
| | KuaiRec | | | KuaiRand | | |
| | Cumulative Reward | Average Reward | Interaction Length | Cumulative Reward | Average Reward | Interaction Length |
| DDPG | $9.2155 \pm 4.05$ | $1.0192 \pm 0.45$ | $9.04 \pm 0.04$ | $1.4232 \pm 0.51$ | $0.3287 \pm 0.12$ | $4.33 \pm 0.03$ |
| PGCR-DDPG | $\mathbf{14.2254 \pm 4.87}$ | $\mathbf{1.5948 \pm 0.55}$ | $\mathbf{8.92 \pm 0.04}$ | $\mathbf{2.0334 \pm 0.65}$ | $\mathbf{0.3657 \pm 0.10}$ | $\mathbf{5.56 \pm 0.03}$ |
| SAC | $10.5235 \pm 3.92$ | $1.1693 \pm 0.44$ | $9.00 \pm 0.10$ | $1.8272 \pm 0.55$ | $0.3500 \pm 0.11$ | $5.22 \pm 0.04$ |
| PGCR-SAC | $\mathbf{15.3726 \pm 4.02}$ | $\mathbf{1.8588 \pm 0.49}$ | $\mathbf{8.27 \pm 0.04}$ | $\mathbf{2.4421 \pm 0.23}$ | $\mathbf{0.4531 \pm 0.05}$ | $\mathbf{5.39 \pm 0.04}$ |
| TD3 | $7.8179 \pm 3.25$ | $0.8610 \pm 0.36$ | $9.09 \pm 0.04$ | $1.5083 \pm 0.40$ | $0.3010 \pm 0.08$ | $5.01 \pm 0.05$ |
| PGCR-TD3 | $\mathbf{14.0021 \pm 4.90}$ | $\mathbf{1.5203 \pm 0.53}$ | $\mathbf{9.21 \pm 0.03}$ | $\mathbf{2.0001 \pm 0.34}$ | $\mathbf{0.3992 \pm 0.07}$ | $\mathbf{5.01 \pm 0.02}$ |

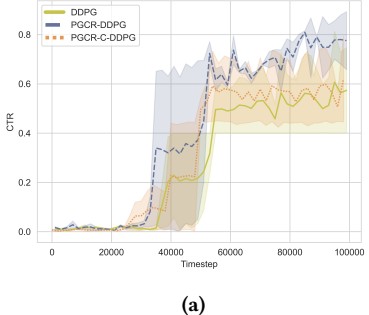
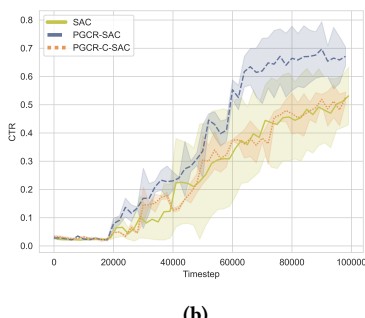
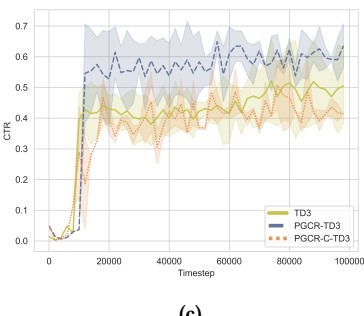

(a)                                    (b)                                    (c)

**Figure 3: Performance comparisons in VirtualTB: (a) DDPG as the backbone, (b) SAC as the backbone, and (c) TD3 as the backbone. Ablation versions with random states are also included in each backbone.**

causal agent with a randomly sampled state. We denote the model without the causal agent as "-C."

Table 2 presents a comparison between the performance of PGCR, the proposed causal state representation learning method, and its variant, PGCR-C, which excludes the causal agent. Across all datasets and reinforcement learning algorithms (DDPG, SAC, TD3), PGCR consistently outperforms PGCR-C in terms of cumulative and average rewards. This highlights the importance and effectiveness of incorporating the causal agent within the PGCR framework, suggesting that the causal state representation significantly enhances the learning process, leading to better policy decisions and improved overall performance.

Regarding interaction length, the differences between PGCR and PGCR-C are generally minor, indicating that the causal agent does not significantly change the duration of interactions but rather improves the quality of decisions during those interactions. The consistent improvements in both cumulative and average rewards across various settings demonstrate that the causal aspect of PGCR

is crucial for achieving optimal performance in reinforcement learning tasks. These results underscore the value of the causal state representation in capturing the underlying structure of the environment, enhancing the algorithm's ability to learn and adapt effectively.

## 4.5 Hyper-parameter Study

In this section, we investigate how the reward balance parameter $\lambda$ in Equation (5) influences the final performance. To account for computational costs, this study is conducted using an online simulation platform, with the results presented in Figure 4. We observe that all three models—PGCR-DDPG, PGCR-SAC, and PGCR-TD3—are highly sensitive to the value of $\lambda$. Each model achieves peak performance in terms of CTR around a $\lambda$ range of 0.1 to 0.2, suggesting that this range is optimal for maximizing the CTR across the models. However, as $\lambda$ increases beyond 0.2, there is a noticeable decline in performance for all models, with PGCR-DDPG experiencing the most significant drop.

**Table 2: Ablation Study**

| | MovieLens-1M | | | Coat | | |
|---|---|---|---|---|---|---|
| | Cumulative Reward | Average Reward | Interaction Length | Cumulative Reward | Average Reward | Interaction Length |
| PGCR-DDPG | 13.0722 ± 3.55 | 4.0587 ± 1.10 | 3.22 ± 0.03 | 19.4281 ± 4.01 | 2.7675 ± 0.57 | 7.02 ± 0.05 |
| PGCR-C-DDPG | 9.9271 ± 4.02 | 3.1022 ± 1.26 | 3.20 ± 0.03 | 17.0237 ± 6.55 | 2.3611 ± 0.91 | 7.21 ± 0.04 |
| PGCR-SAC | 13.4522 ± 3.77 | 4.4544 ± 1.25 | 3.02 ± 0.05 | 20.4272 ± 4.70 | 2.7164 ± 0.63 | 7.52 ± 0.10 |
| PGCR-C-SAC | 11.0238 ± 3.44 | 2.7491 ± 0.86 | 4.01 ± 0.05 | 18.1253 ± 7.02 | 2.5209 ± 0.98 | 7.19 ± 0.03 |
| PGCR-TD3 | 14.1281 ± 5.21 | 3.4375 ± 1.27 | 4.11 ± 0.02 | 19.1192 ± 3.81 | 2.5323 ± 0.50 | 7.55 ± 0.11 |
| PGCR-C-TD3 | 11.0261 ± 4.45 | 3.4349 ± 1.39 | 3.21 ± 0.03 | 17.0221 ± 6.42 | 2.3907 ± 0.91 | 7.12 ± 0.04 |
| | KuaiRec | | | KuaiRand | | |
| | Cumulative Reward | Average Reward | Interaction Length | Cumulative Reward | Average Reward | Interaction Length |
| PGCR-DDPG | 14.2254 ± 4.87 | 1.5948 ± 0.55 | 8.92 ± 0.04 | 2.0334 ± 0.65 | 0.3657 ± 0.10 | 5.56 ± 0.03 |
| PGCR-C-DDPG | 10.4222 ± 4.19 | 1.1087 ± 0.45 | 9.40 ± 0.07 | 1.6410 ± 0.62 | 0.3447 ± 0.13 | 4.76 ± 0.04 |
| PGCR-SAC | 15.3726 ± 4.02 | 1.8588 ± 0.49 | 8.27 ± 0.04 | 2.4421 ± 0.23 | 0.4531 ± 0.05 | 5.39 ± 0.04 |
| PGCR-C-SAC | 11.2890 ± 4.11 | 1.2858 ± 0.47 | 8.78 ± 0.11 | 2.0316 ± 0.41 | 0.3999 ± 0.08 | 5.08 ± 0.05 |
| PGCR-TD3 | 14.0021 ± 4.90 | 1.5203 ± 0.53 | 9.21 ± 0.03 | 2.0001 ± 0.34 | 0.3992 ± 0.07 | 5.01 ± 0.02 |
| PGCR-C-TD3 | 8.1247 ± 3.01 | 0.8793 ± 0.33 | 9.24 ± 0.08 | 1.6218 ± 0.36 | 0.2998 ± 0.07 | 5.41 ± 0.03 |

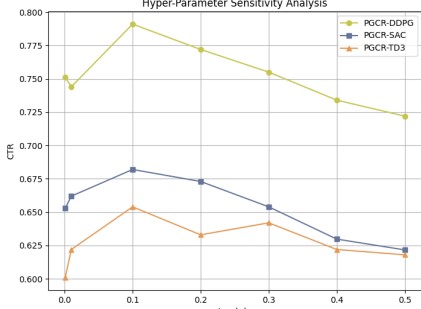

**Figure 4: Hyper Parameter Study in VirtualTB**

## 5 Related Work

**RL-based Recommender Systems** model the recommendation process as a Markov Decision Process (MDP), leveraging deep learning to estimate value functions and handle the high dimensionality of MDPs [19]. Chen et al. [6] proposed InvRec, which uses inverse reinforcement learning to infer rewards directly from user behavior, enhancing policy learning accuracy. Recent efforts have focused on offline RLRS. Wang et al. [25] introduced CDT4Rec, which incorporates a causal mechanism for reward estimation and uses transformer architectures to improve offline RL-based recommendations. Additionally, Chen et al. [4] enhanced this line of research by developing a max-entropy exploration strategy to improve the decision transformer's ability to "stitch" together diverse sequences of user actions, addressing a key limitation in offline RLRS. Gao et al. [9] developed a counterfactual exploration strategy designed to mitigate the Matthew effect, which refers to the disparity in learning from uneven distributions of user data.

**Causal Recommendation.** The recommendation domain has recently seen significant advancements through the integration of causal inference techniques, which help address biases in training data. For example, Zhang et al. [32] tackled the prevalent issue of popularity bias by introducing a causal inference paradigm that adjusts recommendation scores through targeted interventions. Similarly, Li et al. [16] proposed a unified multi-task learning approach to eliminate hidden confounding effects, incorporating a small number of unbiased ratings from a causal perspective. Counterfactual reasoning has also gained traction in recommender systems. Chen et al. [3] developed a causal augmentation technique to enhance exploration in RL-based recommender systems (RLRS) by focusing on causally relevant aspects of user interactions. Wang et al. [26] introduced a method to generate counterfactual user interactions based on a causal view of MDP for data augmentation. In a related vein, Li et al. [17] explored personalized incentive policy learning through an individualized counterfactual perspective. Further studies have focused on the use of causal interventions. Wang et al. [28] proposed CausalInt, a method inspired by causal interventions to address challenges in multi-scenario recommendation. Additionally, He et al. [13] tackled the confounding feature issue in recommendation by leveraging causal intervention techniques. These efforts collectively demonstrate the growing importance of causal inference and intervention in improving recommendation performance and addressing biases.

## 6 Conclusion

In this work, we introduced Policy-Guided Causal Representation (PGCR), a framework designed to enhance state representation learning in offline RL-based recommender systems. By using a causal feature selection policy to isolate the causally relevant components (CRCs) and training an encoder to focus on these components, PGCR effectively improves recommendation performance while mitigating noise and irrelevant features in the state space. Extensive experiments demonstrate the benefits of our approach, confirming its effectiveness in offline RL settings.

For future work, we plan to explore the extension of PGCR to more complex, multi-agent environments where user preferences may dynamically change over time.

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

# Appendix

# A  Definitions in Causality

Here, we briefly introduce some fundamental definitions [21, 22] that are used throughout this paper to present and prove our methodology.

**Definition A.1** (d-Separation [22]). In a Directed Acyclic Graph (DAG) $\mathcal{G}$, a path between two nodes, denoted as $i_n$ and $i_m$, is considered blocked by a set $\mathbf{S}$ if:

(i) Neither $i_n$ nor $i_m$ are included in $\mathbf{S}$, and

(ii) There exists a node $i_k$ on the path such that either:
  (a) $i_k \in \mathbf{S}$ and the connections around $i_k$ follow one of the forms: $i_{k-1} \to i_k \to i_{k+1}$, $i_{k-1} \leftarrow i_k \leftarrow i_{k+1}$, or $i_{k-1} \leftarrow i_k \to i_{k+1}$, or
  (b) $i_k$ is a collider (i.e., $i_{k-1} \to i_k \leftarrow i_{k+1}$), none of its descendants are included in $\mathbf{S}$, and $i_k$ itself is not part of $\mathbf{S}$.

**Definition A.2** (Valid adjustment set [21]). Consider an SCM $\mathcal{M}$ over nodes $\mathbf{V}$ and let $Y \notin \mathrm{PA}_X$ (otherwise we have $p^{\mathcal{M};do(X:=x)}(y) = p^{\mathcal{M}}(y)$). We call a set $\mathbf{Z} \subseteq \mathbf{V} \setminus \{X, Y\}$ a valid adjustment set for the ordered pair $(X, Y)$ if

$$p^{\mathcal{M};do(X:=x)}(y) = \sum_{\mathbf{z}} p^{\mathcal{M}}(y \mid x, \mathbf{z}) p^{\mathcal{M}}(\mathbf{z}).$$

Here, the sum (which could also be an integral) is over the range of $\mathbf{Z}$, that is, over all values $\mathbf{z}$ that $\mathbf{Z}$ can take.

**Definition A.3** (Back-Door Criterion [21]). A set of variables $\mathbf{Z}$ satisfies the back-door criterion relative to an ordered pair of variables $(X_i, X_j)$ in a Directed Acyclic Graph (DAG) $\mathcal{G}$ if:

(i) No node in $\mathbf{Z}$ is a descendant of $X_i$; and
(ii) $\mathbf{Z}$ blocks every path between $X_i$ and $X_j$ that contains an arrow into $X_i$.

Similarly, if $\mathbf{X}$ and $\mathbf{Y}$ are two disjoint subsets of nodes in $\mathcal{G}$, then $\mathbf{Z}$ is said to satisfy the back-door criterion relative to $(\mathbf{X}, \mathbf{Y})$ if it satisfies the criterion relative to any pair $(X_i, X_j)$ such that $X_i \in \mathbf{X}$ and $X_j \in \mathbf{Y}$.

The name "back-door" refers to condition (ii), which requires that only paths with arrows pointing at $X_i$ be blocked; these paths can be viewed as entering $X_i$ through the "back door."

**Definition A.4** (Back-Door Adjustment [21]). If a set of variables $\mathbf{Z}$ satisfies the back-door criterion relative to $(\mathbf{X}, \mathbf{Y})$, then the causal effect of $\mathbf{X}$ on $\mathbf{Y}$ is identifiable and is given by the formula:

$$P^{\mathcal{M};do(X:=x)}(y) = \sum_{\mathbf{z}} P(y \mid x, \mathbf{z}) P(\mathbf{z}).$$

**Definition A.5** (Do-Calculus [21]). Again, consider an SCM over variables $\mathbf{V}$. Let us call an intervention distribution $p^{\mathcal{M};do(X:=x)}(y)$ *identifiable* if it can be computed from the observational distribution and the graph structure. Given a graph $\mathcal{G}$ and disjoint subsets $\mathbf{X}, \mathbf{Y}, \mathbf{Z}, \mathbf{W}$, we have the following:

(1) **Insertion/deletion of observations**:

$$P^{\mathcal{M};do(X:=x)}(y \mid \mathbf{z}, \mathbf{w}) = P^{\mathcal{M};do(X:=x)}(y \mid \mathbf{w})$$

if $\mathbf{Y}$ and $\mathbf{Z}$ are d-separated by $\mathbf{X}, \mathbf{W}$ in a graph where incoming edges into $\mathbf{X}$ have been removed.

(2) **Action/observation exchange**:

$$P^{\mathcal{M};do(X:=x,Z=z)}(y \mid \mathbf{w}) = P^{\mathcal{M};do(X:=x)}(y \mid \mathbf{z}, \mathbf{w})$$

if $\mathbf{Y}$ and $\mathbf{Z}$ are d-separated by $\mathbf{X}, \mathbf{W}$ in a graph where incoming edges into $\mathbf{X}$ and outgoing edges from $\mathbf{Z}$ have been removed.

(3) **Insertion/deletion of actions**:

$$P^{\mathcal{M};do(X:=x,Z=z)}(y \mid \mathbf{w}) = P^{\mathcal{M};do(X:=x)}(y \mid \mathbf{w})$$

if $\mathbf{Y}$ and $\mathbf{Z}$ are d-separated by $\mathbf{X}, \mathbf{W}$ in a graph where incoming edges into $\mathbf{X}$ and $\mathbf{Z}$ (or $\mathbf{W}$) have been removed. Here, $\mathbf{Z}(\mathbf{W})$ is the subset of nodes in $\mathbf{Z}$ that are not ancestors of any node in $\mathbf{W}$ in a graph obtained from $\mathcal{G}$ after removing all edges into $\mathbf{X}$.

# B  Proof of Proposition 1

PROOF. We are given a Structural Causal Model (SCM) that follows the relationships in Equation (7):

$$s_{t+1} = f_P(s_t, a_t, \epsilon_{t+1}), \quad a_t = \pi_t(s_t, \eta_t), \quad r_t = f_R(s_t, a_t), \quad (7)$$

where the state transition function $f_P$ determines the next state $s_{t+1}$ based on the current state $s_t$, action $a_t$, and exogenous noise $\epsilon_{t+1}$. The policy function $\pi_t$ selects the action $a_t$ given the current state $s_t$ and exogenous noise $\eta_t$. The reward function $f_R$ assigns a reward $r_t$ based on the current state $s_t$ and action $a_t$.

We aim to show that $s_t$ satisfies the back-door criterion relative to the pair $(a_t, s_{t+1})$, allowing us to identify the causal effect of $a_t$ on $s_{t+1}$.

**Step 1: Verify the Back-Door Criterion Conditions.** According to Pearl's back-door criterion, for the causal effect of $a_t$ on $s_{t+1}$ to be identifiable, the following conditions must be met:

- No Descendants of $a_t$ in $s_t$: From the given SCM, there are no directed edges from $a_t$ to $s_t$. This means $s_t$ is not a descendant of $a_t$, satisfying the first condition of the back-door criterion.
- Blocking Paths with Arrows into $a_t$: Any back-door path between $a_t$ and $s_{t+1}$ that contains an arrow into $a_t$ must be blocked by $s_t$. In the given SCM, all paths that contain an arrow into $a_t$ are blocked by $s_t$. Specifically, since $s_t \to a_t$, the node $s_t$ acts as a "blocker" for any indirect influence from $a_t$ to $s_{t+1}$ via other variables.

Thus, $s_t$ satisfies the back-door criterion relative to $(a_t, s_{t+1})$.

**Step 2: Identifiability of the Causal Effect.** Since $s_t$ satisfies the back-door criterion, the causal effect of $a_t$ on $s_{t+1}$ is identifiable, meaning we can compute the effect of intervening on $a_t$ on $s_{t+1}$ using observational data.

**Step 3: Derivation of the Intervention Formula.** Using the back-door adjustment, the probability distribution of the state $s_{t+1}$ after an intervention $do(a_t := a_t^\mathcal{I})$ can be computed as:

$$P^{\mathcal{M};do(a_t:=a_t^\mathcal{I})}(s_{t+1})$$

$$= \sum_{s_t} \int_{\epsilon_{t+1}} P(s_{t+1} \mid do(a_t), s_t, \epsilon_{t+1}) P(\epsilon_{t+1}) P(s_t \mid do(a_t)) \, d\epsilon_{t+1}.$$

Since $P(s_t \mid do(a_t)) = P(s_t)$, we simplify the expression:

$$P^{\mathcal{M};do(a_t:=a_t^{\mathcal{I}})}(s_{t+1}) = \sum_{s_t} \int_{\epsilon_{t+1}} P(s_{t+1} \mid s_t, a_t^{\mathcal{I}}, \epsilon_{t+1}) \, P(\epsilon_{t+1}) \, P(s_t) \, d\epsilon_{t+1}.$$

Finally, the expression simplifies to:

$$P^{\mathcal{M};do(a_t:=a_t^{\mathcal{I}})}(s_{t+1}) = \mathbb{E}_{s_t, \epsilon_{t+1}} \left[ P(s_{t+1} \mid s_t, a_t^{\mathcal{I}}, \epsilon_{t+1}) \right].$$

This equation shows that the causal effect of $a_t$ on $s_{t+1}$ is identifiable through the expected value of the conditional probability distribution, considering the distribution of $s_t$ and the exogenous noise $\epsilon_{t+1}$. □

## C  Proof of Proposition 2

PROOF. We want to show that for any $s, s^\circ \in \mathcal{S}$ such that $\phi(s) = \phi(s^\circ)$, the optimal action-value function satisfies:

$$Q^*(s, a) = Q^*(s^\circ, a).$$

**Step 1: Bellman Equation for the Action-Value Function.** The Bellman equation for the optimal action-value function $Q^*(s, a)$ is:

$$Q^*(s, a) = \mathbb{E} \left[ r(s, a) + \gamma \max_{a'} Q^*(s', a') \mid s, a \right].$$

Since we are given that $r(s, a) \perp\!\!\!\perp s \mid z = \phi(s), a$, the reward $r(s, a)$ depends only on the latent state $z = \phi(s)$, and not on the full state $s$. Therefore, the reward term in the Bellman equation can be rewritten as:

$$Q^*(s, a) = \mathbb{E} \left[ r(z, a) + \gamma \max_{a'} Q^*(s', a') \mid z = \phi(s), a \right].$$

**Step 2: Rewards Depend on Latent State.** Because the reward $r(s, a)$ depends only on the latent state $z = \phi(s)$, we have:

$$\mathbb{E} \left[ r(s, a) \mid z \right] = \mathbb{E} \left[ r(s^\circ, a) \mid z \right] \quad \text{whenever} \quad \phi(s) = \phi(s^\circ).$$

Thus, the expected reward for the states $s$ and $s^\circ$ is identical if the latent state representations are the same.

**Step 3: Transition Dynamics Depend on Latent State.** We are given that for all $s, s^\circ \in \mathcal{S}$ such that $\phi(s) = \phi(s^\circ)$, the probability distribution of the next latent state satisfies:

$$p(\phi(s') \mid s) = p(\phi(s') \mid s^\circ).$$

This means that the transition dynamics between latent states are identical for the states $s$ and $s^\circ$. Since the transition dynamics depend only on the latent state, the distribution of future latent states $z' = \phi(s')$ is the same whether we start from $s$ or $s^\circ$.

**Step 4: Expectation Over Next State.** The next state $s'$ depends on the current state $s$ and action $a$, but the latent state dynamics depend only on the latent representation $z = \phi(s)$. Therefore, the expectation over the future action-value function $Q^*(s', a')$ in the Bellman equation depends only on the latent state $z = \phi(s)$. Thus, we have:

$$\mathbb{E} \left[ \max_{a'} Q^*(s', a') \mid z = \phi(s), a \right] = \mathbb{E} \left[ \max_{a'} Q^*(s', a') \mid z = \phi(s^\circ), a \right].$$

Since both the reward term and the expected value of the next action-value function depend only on the latent state $z = \phi(s)$, we conclude that:

$$Q^*(s, a) = Q^*(s^\circ, a) \quad \text{whenever} \quad \phi(s) = \phi(s^\circ).$$

Thus, the optimal action-value function depends only on the latent state $z = \phi(s)$, and not on the full state $s$, which completes the proof. □

