# OpenReview forum: "Policy-Guided Causal State Representation  for Offline Reinforcement Learning Recommendation"
_ACM.org/TheWebConf/2025/Conference — WWW 2025 Oral_

### Official Review · Reviewer_YoFb · 2024-11-15

**Novelty:** 4
**Technical Quality:** 4

**Review:**

**Summary**

This paper introduces PGCR (Policy-Guided Causal Representation), a novel two-stage framework for enhancing state representation learning in offline reinforcement learning-based recommender systems. PGCR operates in two stages. In the first stage, it learns a policy to identify Causally Relevant Components (CRCs) using a reward function based on the Wasserstein distance. In the second stage, it trains an encoder for state representation learning, guided by the policy learned in the previous stage. The paper provides theoretical analysis of the proposed method and demonstrates performance improvements through experiments on various datasets.

**Strengths**

1.	Solid theoretical analysis of causal effect identifiability
2.	Novel application of Wasserstein distance for the reward function
3.	Demonstrated versatility through successful integration with various existing RL frameworks

**Weakness**

1.	Inadequate Problem Motivation
-	The problem statement in the introduction ("However, raw state representations are often complex and may contain components that are not causally relevant to the reward") is too general and abstract. The paper lacks concrete examples or empirical evidence demonstrating the severity of this issue and fails to analyze why existing approaches are insufficient.
-	The problem of "missing transitions in offline data" mentioned in the abstract is not adequately explained in terms of its importance. The causal relationship between the proposed method and this problem's resolution is unclear, and the paper lacks specific scenarios illustrating how this issue manifests in real recommendation systems.
2.	Limited Evaluation Metrics for Recommendation Systems
-	The paper relies solely on cumulative reward, average reward, and interaction length for evaluation, rather than using traditional and well-established recommendation metrics (such as Recall, NDCG, MRR). This makes it difficult to assess whether the proposed method actually improves recommendation quality or user satisfaction.
-	If traditional metrics are not applicable, the study would benefit from A/B testing or user studies to validate that the proposed method leads to actual improvements in user satisfaction.
-	Additionally, if the experimental setting is modified, it would be appropriate to include offline RLRS-related baselines (such as CDT4Rec mentioned in line 857).
3.	Experimental Setting Issues
-	The data splitting methodology is not clearly described. The paper fails to specify the train/validation/test split ratios and whether the splitting considers temporal ordering or uses random partitioning, significantly hampering reproducibility.
-	Section 4.2 merely lists hyperparameter values without explaining the selection process. It's unclear which evaluation metrics guided the tuning process and what optimization methodology (e.g., grid search, random search) was employed, lacking transparency in the optimization process.
-	The choice of GRU as the state encoder for offline datasets ("When converting them into the RL environments, we use the GRU as the state encoder for those offline datasets") lacks justification. The paper should either cite relevant prior work if following established practices or explain the reasoning behind this architectural choice if it's an original decision.

**Questions:**

1.	Could you provide concrete examples of how non-causally relevant components in state representations negatively impact recommendation performance? Specifically, could you demonstrate through empirical analysis or examples how these components interfere with learning and quantify their impact on recommendation quality? This seems crucial for justifying the need for causal feature selection.
2.	Why weren't traditional recommendation metrics included in the evaluation? How can we assess the practical effectiveness of PGCR in real-world recommendation scenarios without standard metrics like NDCG, Recall, or MRR? Could you provide additional evaluation using these conventional metrics to facilitate comparison with existing recommendation systems and validate the method's practical utility?
3.	Given that the paper focuses on offline RLRS, why weren't recent offline RLRS methods included in the baseline comparisons? This comparison seems essential for understanding the actual contribution of the causal representation learning approach in the context of recommendation systems.

**Reviewer Confidence:**

2: The reviewer is willing to defend the evaluation, but it is likely that the reviewer did not understand parts of the paper

**Scope:**

3: The work is somewhat relevant to the Web and to the track, and is of narrow interest to a sub-community

---

### Official Review · Reviewer_q1Yx · 2024-11-16

**Novelty:** 4
**Technical Quality:** 6

**Review:**

The paper proposes PGCR (Policy-Guided Causal Representation), a novel two-stage framework for offline reinforcement learning-based recommender systems. The first stage uses a causal feature selection policy to identify and retain causally relevant components of states while modifying irrelevant ones. The second stage trains an encoder to learn state representations focused on these causal components. The approach uses Wasserstein distance to measure causal effects and guides the learning process through a specialized reward function.

**Questions:**

1. What is “Your proposition looks logically sound, and the structure is clear. To improve consistency, here’s a refined version of the proposition with a small clarification at the end:”, it may be comments from authors.
2. I am a little confused how to select causal feature in section 3.2, can authors briefly talk about it?
3. It seems authors only used causal inference techniques like SCM to improve the state in RL, can authors talk about is there any technical improvement rather than combine existing techniques? My main concern is in this technical novelty.

**Reviewer Confidence:**

4: The reviewer is certain that the evaluation is correct and very familiar with the relevant literature

**Scope:**

4: The work is relevant to the Web and to the track, and is of broad interest to the community

---

### Official Review · Reviewer_kUr2 · 2024-11-25

**Novelty:** 5
**Technical Quality:** 5

**Review:**

This paper addresses a critical challenge in applying offline Reinforcement Learning (RL) to recommender systems: how to identify and isolate causally relevant features to improve decision-making quality. The authors propose a causal feature selection policy that uses causal interventions to eliminate causally irrelevant components (CIRCs) and integrate it into a state representation learning framework. The method utilizes an encoder to map states into a latent space that focuses on key causally relevant components (CRCs), thereby enhancing the recommendation system's accuracy.

### **Quality**

The paper is well-structured, with a rigorous methodology supported by appropriate mathematical formulations. Key results are derived in detail, and the application of causal inference principles within the offline RL framework is theoretically grounded. Overall, the research quality is high.

### **Clarity**

The paper is clearly written, with thorough explanations of the formulas, algorithms, and key concepts. The transitions between sections are smooth, and the detailed descriptions help readers understand the proposed method. Both the algorithms and the formulas are presented in an easy-to-understand manner, making the paper accessible to the intended audience.

### **Originality**

This work directly applies causal feature selection to optimize state representations in the offline RL framework, introducing a novel approach to improve the decision quality of recommender systems. The innovative use of causal interventions and Wasserstein distance to quantify and optimize the causally relevant components in state representations adds to its originality.

### **Significance**

The paper is of significant practical importance in the domain of recommender systems. By focusing on causally relevant features, it helps to more accurately extract the key factors influencing user satisfaction, reducing interference from irrelevant components, and thus optimizing recommendation performance.

### **Pros**

1. **Sound Motivation**: The authors clearly highlight the noise issues in state representations within offline RL and propose an effective solution.
2. **Detailed Algorithmic Steps**: Algorithms 1 and 2 are well-described and facilitate understanding of the specific implementation details.
3. **Strong Theoretical Foundation**: The propositions and methods in the paper are supported by detailed theoretical proofs, providing a solid foundation for the research results.

### **Cons**

1. The absence of practical illustrations limits the reader's ability to grasp the real-world applicability of the method.
1. The selection of Wasserstein distance lacks sufficient explanation, leaving a gap in the justification for critical design choices.

**Questions:**

1. **Could you provide a concrete example illustrating how $a_t^I$ is set and how it leads to the generation of $s^I$?**

2. **The paper mentions the use of Wasserstein distance to measure causal relevance, but it does not fully explain the advantage of this choice.** Could you clarify why Wasserstein distance is chosen in this context over other commonly used distance metrics (e.g., KL divergence )?

**Reviewer Confidence:**

2: The reviewer is willing to defend the evaluation, but it is likely that the reviewer did not understand parts of the paper

**Scope:**

3: The work is somewhat relevant to the Web and to the track, and is of narrow interest to a sub-community

---

### Official Review · Reviewer_Un9T · 2024-11-29

**Novelty:** 7
**Technical Quality:** 7

**Review:**

## Review

This paper proposes a novel two-stage framework for causal feature selection and state representation learning in offline reinforcement learning recommendations (RLRS). The proposed framework, Policy-Guided Causal Representation (PGCR), provides a well-structured approach to improving state representation learning for RLRS by focusing on causally relevant features that aim to isolate critical state components for decision-making. It thus effectively combines causal modeling with reinforcement learning to address critical challenges in these settings.

The motivation and problem formulation are clear. The methodology is conceptually robust, detailed, and is supported by theoretical analysis that demonstrates the identifiability of causal effects from interventions, as well as well-designed algorithms for learning the policies. As such, the work's significance lies in tackling key limitations and challenges of RLRS and in the improved recommendation performance, as demonstrated through extensive experiments with standard RL algorithms.

The work is original in its integration of causal inference through interventions to improve state representations, leveraging Wasserstein distance to measure the impact of interventions on reward distributions, and adapting these techniques specifically for static datasets in offline RL scenarios. While it builds on existing frameworks like bisimulation metrics and encoder-based latent representations, the approach of combining causal selection policies with RLRS is innovative.

### Pros
- The two-stage framework (causal feature selection + encoder training) is logically sound and theoretically justified.
- Use of Wasserstein distance for causal effect quantification and its integration to learn reward policies is novel.
- Theoretical analysis of identifiability of causal effects and derivation of optimal policies based on latent state representations.
- Thorough empirical validation with extensive experiments on multiple static datasets and standard RL algorithms, as well as on an online simulation platform.
- The focus on offline RLRS makes it applicable to many domains where real-time interactions are impractical.
- The paper is well-structured and builds on well-established frameworks and mathematical notations.

### Cons
- On Page 3, the phrase:
  > Your proposition looks logically sound, and the structure is clear. To improve consistency, here’s a refined version of the proposition with a small clarification at the end:

  appears, making it hard to estimate to what extent the authors used GenAI for the manuscript, as they do not disclose their use in the paper (e.g., generate new content, merely edit and improve quality, etc.). I refer to the [ACM Policy on Authorship](https://www.acm.org/publications/policies/frequently-asked-questions).

- The authors mention CIDS as a concurrent work and discuss theoretical differences between PGCR and CIDS. The authors argue that CIDS is tailored for online RLRS; however, this justification becomes inconsistent given their own use of an online RLRS environment for experiments.

**Questions:**

## Questions

- I am interested to know why was the Wasserstein distance chosen as the primary metric for quantifying the similarity between observational and intervened reward distributions?
- The paper assumes no unobserved confounders. However, in practical recommendation systems, causal relationships are often confounded by unobserved variables or spurious correlations (e.g., user mood, external trends). How could the proposed method be extended to such cases, i.e. how would the authors account for potential confounders, and to cases where the causal graph is partially or incorrectly specified?
- Does the frameowrk consider interpretability, i.e. latent space representations and interpretability of CRCs?

**Reviewer Confidence:**

3: The reviewer is confident but not certain that the evaluation is correct

**Scope:**

3: The work is somewhat relevant to the Web and to the track, and is of narrow interest to a sub-community